# Pretreatment Modified Albumin–Bilirubin Grade Is an Important Predictive Factor Associated with the Therapeutic Response and the Continuation of Atezolizumab plus Bevacizumab Combination Therapy for Patients with Unresectable Hepatocellular Carcinoma

**Takashi Tanaka** *[ID]**, Kazuhide Takata** [ID]**, Keiji Yokoyama, Hiromi Fukuda, Ryo Yamauchi** [ID]**, Atsushi Fukunaga, Satoshi Shakado, Shotaro Sakisaka and Fumihito Hirai**

Department of Gastroenterology and Medicine, Faculty of Medicine, Fukuoka University, Nanakuma 7-45-1, Fukuoka 814-0180, Japan; edihuzak_t@yahoo.co.jp (K.T.); yokotin@fukuoka-u.ac.jp (K.Y.); hiromi.m.0928@gmail.com (H.F.); r_christinue2017@yahoo.co.jp (R.Y.); afukufuku628@outlook.jp (A.F.); shakado@cis.fukuoka-u.ac.jp (S.S.); sakisaka@fukuoka-u.ac.jp (S.S.); fuhirai@cis.fukuoka-u.ac.jp (F.H.)
* Correspondence: tanakatakashi329@gmail.com; Tel.: +81-92-801-1011; Fax: +81-92-874-2663

**Abstract:** Background: Atezolizumab plus bevacizumab (ATZ + BV) treatment is recommended as the first-line systemic therapy for patients with unresectable hepatocellular carcinoma (u-HCC). This study aimed to investigate the predictive factors of therapeutic response and the continuation of ATZ + BV treatment for u-HCC in a real-world setting. Methods: This retrospective study was conducted between January 2021 and April 2022. Twenty-eight patients with u-HCC, who were treated with ATZ + BV, were assessed for their treatment response, continuation, and adverse events (AEs). Results: Among the 28 patients, 24 were evaluated at the first imaging. The objective response rate (ORR) was 29.2% ($n = 7$), and 54.2% ($n = 13$) on the response evaluation criteria in solid tumors (RECIST 1.1) and in the modified RECIST (mRECIST) guidelines, respectively. Comparing the objective response (OR) group ($n = 13$) and the non-OR group ($n = 11$), the modified albumin–bilirubin (mALBI) grades 1 and 2a were found to be significant predictive factors for OR ($p = 0.021$) in the mRECIST guidelines. Among the 28 patients, 17 discontinued their treatment due to AEs. Comparing the treatment continuation ($n = 11$) and discontinuation groups ($n = 17$), a Child–Pugh score of five points ($p = 0.009$) and mALBI grades 1 and 2a ($p = 0.020$) were predictive factors with significant differences. Conclusions: Pretreatment mALBI grades 1 and 2a were the important predictive factors associated with the therapeutic response and the therapeutic continuation of ATZ + BV for patients with u-HCC.

**Keywords:** hepatocellular carcinoma; atezolizumab; bevacizumab; immune checkpoint inhibitors; adverse events



## 1. Introduction

Hepatocellular carcinoma (HCC) is one of the leading causes of cancer-related deaths worldwide [1]. Recently, systemic therapeutic strategies have been developed for patients with unresectable HCC (u-HCC) [2,3]. Sorafenib was developed as the initial first-line molecular target agent (MTA) in 2009 [4], and, in Japan, lenvatinib was granted as an additional first-line MTA for u-HCC in 2018 [5]. Regarding second-line MTA treatments, regorafenib [6], ramucirumab [7], and cabozantinib [8] were approved in Japan in 2017, 2019, and 2020, respectively. Recently, immunotherapy appears to be a promising therapeutic approach for HCC, and combining immunotherapy with other treatment modalities, such as monoclonal antibodies, tyrosine kinase inhibitors, or local therapies, can increase the overall response rate and survival rate [9]. Furthermore, atezolizumab plus bevacizumab

(ATZ + BV) was recently approved for use as a first-line systemic therapy for patients with u-HCC, according to the American Society of Clinical Oncology guidelines [10], based on the IMbrave150 trial results [11]. Atezolizumab (ATZ) is a humanized monoclonal antibody that is programmed to cell death-ligand 1 (PD-L1), which blocks the binding of PD-L1 to programmed cell death-1 (PD-1) and restores anti-cancer immunity [12]. Bevacizumab (BV) targets the vascular endothelial growth factor for angiogenesis and tumor growth [13,14]. In the IMbrave150 trial, the ATZ + BV treatment maintained the patients' quality of life and improved the survival benefits more than sorafenib was able to [10]. The therapeutic potential of ATZ + BV for u-HCC in clinical practice has also been reported in several recent studies [15–21]. However, HCC patients with a history of MTA therapy and those with Child–Pugh class B cirrhosis were excluded from the IMbrave150 trial. Consequently, recent reports regarding the safety and efficacy of ATZ + BV in these patients are inadequate. Moreover, reliable predictive markers of therapeutic response are necessary for patients with u-HCC, who are treated by ATZ + BV therapy in a real-world setting [22]. As a result, these factors should play an important role in selecting the appropriate treatment for u-HCC patients.

This study aimed to assess the efficacy and safety of ATZ + BV treatment for patients with u-HCC from the viewpoint of their clinical features and to evaluate the pretreatment factors related to the therapeutic response and continuation of ATZ + BV treatment in a real-world setting.

## 2. Materials and Methods

### 2.1. Patient Population

This single-center study retrospectively analyzed 28 patients with u-HCC, who were treated with ATZ + BV between January 2021 and April 2022. The study was approved by the hospital's institutional review board (approval number: H21-10-0002) and conducted in accordance with the tenets of the Declaration of Helsinki. Written informed consent was obtained from all patients prior to treatment. HCC was diagnosed by radiological imaging, using contrast-enhanced computed tomography (CT) or magnetic resonance imaging (MRI), combined with serum tests, particularly tumor markers such as alpha-fetoprotein (AFP) and des-gamma carboxy prothrombin (DCP) [23]. All patients were at least 20 years old. Patients with pregnancy, poor liver function (as indicated by a classification of Child–Pugh class C), poor Eastern Cooperative Oncology Group (ECOG) performance status, a known history of autoimmune disease, and those judged as inappropriate by the attending physician were excluded from the study.

### 2.2. Study Design

ATZ (1200 mg; Chugai Co., Ltd., Tokyo, Japan) and BV (15 mg/kg; Chugai Co., Ltd.) were administered intravenously every 3 weeks. Clinical data, such as age; sex; HCC etiology; ECOG performance status; hepatic functional reserves, including Child–Pugh score; albumin–bilirubin (ALBI) score; modified ALBI (mALBI) grade; naïve or recurrent HCC; Barcelona Clinical Liver Cancer group (BCLC) stage; tumor markers; AFP and DCP levels; size and number of tumors; vascular invasion; metastasis to other organs; and neutrophil-to-lymphocyte ratio (NLR) were evaluated. The ALBI grade was defined based on the serum albumin and total bilirubin values, using a specific formula and was classified into grades 1–3 [24]. In the mALBI grading, ALBI grade 2 was divided into 2a and 2b, using an ALBI score cut-off value of $-2.270$ [25]. The first assessment of therapeutic response was performed using dynamic CT results, obtained approximately 6 to 9 weeks after the introduction of ATZ + BV, and additional dynamic CT examinations were performed as required, based on the patient's condition, every 6 to 9 weeks. Therapeutic responses were determined using both the Response Evaluation Criteria in Solid Tumors (RECIST) version 1.1 [26], and modified RECIST (mRECIST) guidelines [27]. The objective response rate (ORR) was assessed as the complete response (CR) plus partial response (PR). The disease control rate (DCR) was assessed as objective response (OR) plus stable disease (SD). Therapeutic response was diagnosed by expert radiologists in our institute, based on a previous study [28].

The safety profile of the combination was evaluated after treatment initiation. Treatment was discontinued when unacceptable adverse events (AEs) or progressive disease (PD) was observed on imaging. Additionally, combination therapy was interrupted if patients developed grade 3 or higher AEs or unacceptable AEs. AEs and grades were defined according to the National Cancer Institute Common Terminology Criteria for Adverse Events (CTCAE) version 5.0, and American Society of Clinical Oncology Clinical Practice Guidelines [29].

### 2.3. Study Endpoint

The primary endpoint was to evaluate the predictive factors for the therapeutic response to ATZ + BV treatment. Patients were divided into two groups according to the mRECIST guidelines, as follows: an OR group that included patients with CR and PR and a non-objective response (non-OR) group that included patients with SD and PD. The two groups were compared using statistical analysis. The secondary endpoint was to evaluate the factors associated with the treatment continuation of ATZ + BV therapy due to AEs and to assess the contents and grade of AEs. Patients were further divided into two groups, continuous and discontinuous, and the pretreatment factors were compared using statistical analysis.

### 2.4. Statistical Analysis

Continuous variables were analyzed using the Mann–Whitney U-test. Categorical variables were analyzed using Fisher's exact test. Overall survival and progression-free survival rates were analyzed using the Kaplan–Meier technique, and the differences in curves were assessed using the log-rank test. In all analyses, a *p*-value of <0.05 was considered as statistically significant. All statistical analyses were performed using JMP software for Windows version 14.3 (SAS Institute, Cary, NC, USA).

## 3. Results

### 3.1. Patient Characteristics

The baseline characteristics of the patients enrolled in this study are shown in Table 1. The median patient age was 73.5 (range, 56–89) years, and the cohort included 22 men and six women. The etiologies of liver disease were as follows: one case of hepatitis B virus infection, 12 cases of hepatitis C virus infection, and 15 cases of non-viral etiology. Additionally, 14, 7, and 7 patients had baseline Child–Pugh scores of five, six, and seven points, respectively. All seven patients who had a Child–Pugh class B at treatment initiation, had a Child–Pugh class A at the time of the treatment decision for u-HCC. The number of patients with mALBI grades 1, 2a, 2b, and 3 was 8, 7, 12, and 1, respectively. The number of patients with BCLC stages A, B, and C, was 3, 10, and 15, respectively. Among the 28 patients, 13 had naïve HCC. Additionally, 23 patients underwent ATZ + BV treatment as first-line systemic chemotherapy, and five had an MTA history before the ATZ + BV treatment.

**Table 1.** The baseline characteristics of patients.

| Characteristics | | Values (*n* = 28) |
|---|---|---|
| Age (years) † | | 73.5 (56, 89) |
| Sex * | | |
| | Male/Female | 22 (79)/6 (21) |
| Etiology * | | |
| | HBV/HCV/non-viral | 1 (4)/12 (43)/15 (53) |
| ECOG PS * | | |
| | 0/1/2 | 25 (89)/2 (7)/1 (4) |
| Child–Pugh score * | | |
| | 5/6/7 | 14 (50)/7 (25)/7 (25) |
| mALBI grade * | | |
| | 1/2a/2b/3 | 8 (28)/7 (25)/12 (43)/1 (4) |
| Naïve HCC * | | |
| | Yes/No | 13 (46)/15 (54) |
| BCLC stage * | | |
| | A/B/C | 3 (11)/10 (36)/15 (53) |

**Table 1.** *Cont.*

| Characteristics | Values (*n* = 28) |
|---|---|
| Size of tumor (cm) † | 5.3 (0.5, 18.5) |
| Number of tumor † | 2 (1, 50) |
| Portal vain invasion * | |
| Yes/No | 7 (25)/21 (75) |
| Metastasis to other organs * | |
| Yes/No | 9 (32)/19 (68) |
| AFP (ng/mL) † | 310 (0.5, 769,600) |
| DCP (mAU/mL) † | 547 (14, 136,617) |
| NLR † | 3.13 (1.19, 25.7) |

† Median (minimum, maximum). * Number of the patients (%). HBV—hepatitis B virus; HCV—hepatitis C virus; non-viral, alcoholic, non-alcoholic steatohepatitis and autoimmune hepatitis; ECOG—Eastern Cooperative Oncology Group; PS—performance status; mALBI—modified albumin–bilirubin; HCC—hepatocellular carcinoma; BCLC—Barcelona Clinic Liver Cancer; AFP—alpha-fetoprotein; DCP—des-gamma-carboxy prothrombin; NLR—neutrophil–to–lymphocyte ratio.

### 3.2. Therapeutic Efficacy and Factors Associated with Therapeutic Response

We analyzed the therapeutic responses at 6 or 9 weeks, at the first imaging examination after initiating the ATZ + BV therapy, using a dynamic CT or MRI, according to the RECIST 1.1 and mRECIST guidelines. Among the 28 patients in this study, 24 were evaluated at the first imaging examination, and of those none (0%), 6 (25%), 16 (66.7%), and 2 (8.3%) showed CR, PR, SD, and PD, respectively, based on the RECIST 1.1 guidelines, while none (0%), 13 (54.2%), 9 (37.5%), and 2 (8.3%) showed CR, PR, SD, and PD, respectively, based on the mRECIST guidelines. The frequencies of patients with CR and PR (i.e., ORR) and CR, PR, and SD (i.e., DCR) were 25% and 91.7%, respectively, based on the RECIST 1.1 guidelines, and 54.2% and 91.7%, respectively, based on the mRECIST guidelines (Figure 1a). The best responses among the 24 patients were as follows: 1 (4.2%), 6 (25%), 15 (62.5%), and 2 (8.3%) experienced CR, PR, SD, and PD, respectively, based on the RECIST 1.1 guidelines, and 1 (4.2%), 12 (50.0%), 9 (37.5%), and 2 (8.3%) experienced CR, PR, SD, and PD, respectively, based on the mRECIST guidelines. The ORR and DCR of the best response were 29.2% and 91.7%, respectively, based on the RECIST 1.1 guidelines, and 54.2% and 91.7%, respectively, based on the mRECIST guidelines (Figure 1b). The factors associated with comparing the therapeutic response (in the mRECIST guidelines) between the OR group (*n* = 13) and the non-OR group (*n* = 11) are shown in Table 2. Consequently, mALBI grades 1 and 2a showed statistically significant differences in the therapeutic response between both groups (*p* = 0.021).

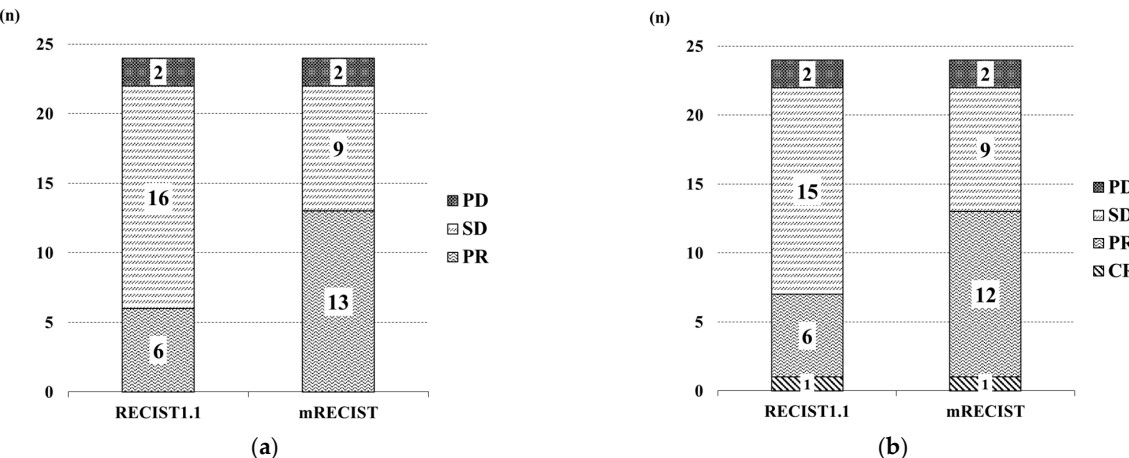

**Figure 1.** Therapeutic response of atezolizumab plus bevacizumab. (**a**) Assessment of the therapeutic response at first CT or MRI imaging, based on Response Evaluation Criteria in Solid Tumors (RECIST) 1.1 and modified RECIST (mRECIST) guidelines. (**b**) Assessment of the best overall re-sponse, based on RECIST 1.1 and modified RECIST guidelines. CR—complete response; PR—partial response; SD—stable disease; PD—progressive disease.

**Table 2.** Predictive factors between OR and non-OR groups.

| Characteristics | OR (*n* = 13) | Non-OR (*n* = 11) | *p* Value |
|---|---|---|---|
| Age (years) | 73 (59, 84) | 72 (56, 89) | 0.331 † |
| Sex: Male/Female | 9/4 | 9/2 | 0.410 * |
| Etiology: HBV and HCV/Non-viral | 5/8 | 7/4 | 0.207 * |
| ECOG PS: 0/≥1 | 13/0 | 9/2 | 0.199 * |
| Naïve/recurrence HCC | 6/7 | 4/7 | 0.473 * |
| Child Pugh score: 5/≥6 | 9/4 | 4/7 | 0.115 * |
| mALBI grade: 1 and 2a/2b and 3 | 10/3 | 3/8 | 0.021 * |
| BCLC stage: A and B/C | 4/9 | 7/4 | 0.115 * |
| Portal vein invasion: Yes/No | 5/8 | 2/9 | 0.264 * |
| Metastasis other organs: Yes/No | 5/8 | 2/9 | 0.264 * |
| AFP (ng/mL) | 261 (0.7, 14,705) | 764 (9.1, 769,600) | 0.170 † |
| DCP (mAU/mL) | 403 (14, 136,617) | 447 (29, 33,381) | 0.172 † |
| NLR | 3.00 (1.19, 6.52) | 3.10 (1.41, 25.7) | 0.214 † |

Values are presented median (minimum, maximum) or number. † *p*-value with continuous variables were obtained by Mann–Whitney U test. * *p*-value with categorical carriables were obtained by the Fisher's exact test. OR—objective response; HBV—hepatitis B virus; HCV—hepatitis C virus; non-viral, alcoholic, non-alcoholic steatohepatitis and autoimmune hepatitis.

### 3.3. Overall Survival and Progression-Free Survival

We analyzed the overall survival and the progression-free survival rates in this study population. Kaplan–Meier curves for the overall survival and progression-free survival rates are shown in Figure 2. The median overall survival rate was not evaluated (range, 52–472), and the overall survival rate at 180 days was 85% (Figure 2a). The overall survival rates between the mALBI grade 1 plus 2a group and the mALBI grade 2b plus 3 group were not significantly different (*p* = 0.059, log-rank) (Figure 2b). The median progression-free survival rate was 316 days (range, 52–472), and the progression-free survival rate at 180 days was 81.4% (Figure 2c). The progression-free survival rates between the mALBI grade 1 plus 2a group and the mALBI grade 2b plus 3 group were not significantly different (*p* = 0.224, log-rank) (Figure 2d).

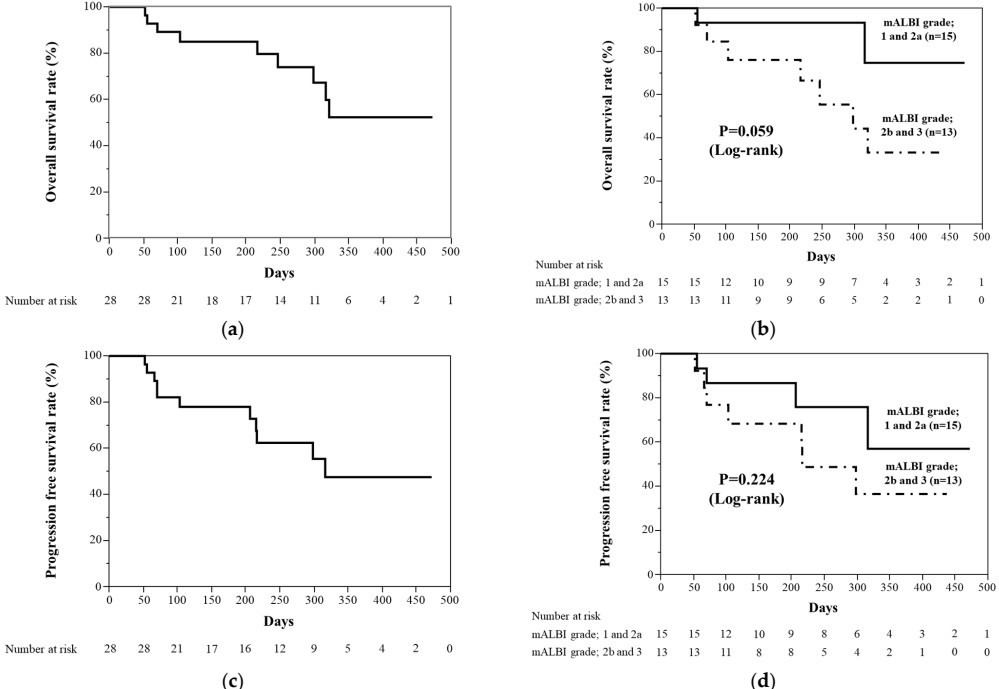

**Figure 2.** Kaplan–Meier curves for overall survival rate and progression-free survival rate. (**a**) Overall survival rate. Median overall survival rate was not evaluated. (**b**) Overall survival rates between the mALBI grade 1 plus 2a group and mALBI grade 2b plus 3 group were not significantly different (*p* = 0.059). (**c**) Progression-free survival rate. Median progression-free survival was 316 days. (**d**) Progression-free survival rates between mALBI grade 1 plus 2a group and mALBI grade 2b plus 3 group were not significantly different (*p* = 0.224).

*3.4. Factors Associated with Treatment Discontinuation*

Among the 28 patients, 17 discontinued the ATZ + BV treatment because of AEs. We investigated the factors associated with the treatment continuation between the treatment continuation ($n$ = 11) and treatment discontinuation groups ($n$ = 17) (Table 3). Consequently, a Child–Pugh score of five points ($p$ = 0.009) and the mALBI grades 1 and 2a ($p$ = 0.020) had a statistically different association with the continuation of the ATZ + BV treatment between both groups. Figure 3 shows the clinical course of all patients, who were divided into the mALBI grade 1 plus 2a ($n$ = 15) and 2b plus 3 groups ($n$ = 13). The course of the therapeutic response and the treatment continuation in the mALBI grade 1 plus 2a group had a larger population of OR and longer treatment continuation than in the mALBI grade 2b plus 3 group.

**Table 3.** Predictive factors between treatment continuation and discontinuation groups.

| Characteristics | Continuation ($n$ = 11) | Discontinuation ($n$ = 17) | $p$ Value |
|---|---|---|---|
| Age (years) | 71 (58, 87) | 76 (56, 89) | 0.085 † |
| Sex: Male/Female | 10/1 | 12/5 | 0.160 * |
| Etiology: HBV and HCV/non-viral | 5/6 | 8/9 | 0.479 * |
| ECOG PS: 0/≥1 | 11/0 | 14/3 | 0.171 * |
| Naïve/recurrence HCC | 3/8 | 10/7 | 0.106 * |
| Child–Pugh score: 5/≥6 | 9/2 | 5/12 | 0.009 * |
| mALBI grade: 1 and 2a/2b and 3 | 9/2 | 6/11 | 0.020 * |
| BCLC stage: A and B/C | 3/8 | 10/7 | 0.106 * |
| Portal vein invasion: Yes/No | 4/7 | 3/14 | 0.250 * |
| Metastasis other organs: Yes/No | 3/8 | 5/12 | 0.624 * |
| AFP (ng/mL) | 40 (0.7, 17,483) | 590 (1.4, 769,600) | 0.166 † |
| DCP (mAU/mL) | 419 (14, 28,256) | 651 (41, 136,617) | 0.076 † |
| NLR | 3.18 (1.63, 6.52) | 3.07 (1.19, 25.7) | 0.101 † |

Values are presented as median (minimum, maximum) or number. † $p$-value with continuous variables were obtained by Mann–Whitney U test. * $p$-value with categorical carriables were obtained by the Fisher's exact test. HBV—hepatitis B virus; HCV—hepatitis C virus; non-viral, alcoholic, non-alcoholic steatohepatitis and autoimmune hepatitis; ECOG—Eastern Cooperative Oncology Group; PS—performance Status; HCC—hepatocellular carcinoma; mALBI—modified albumin-bilirubin; BCLC—Barcelona Clinic Liver Cancer; AFP—alpha-fetoprotein; DCP—des-gamma-carboxy prothrombin; NLR—neutrophil–to–lymphocyte ratio.

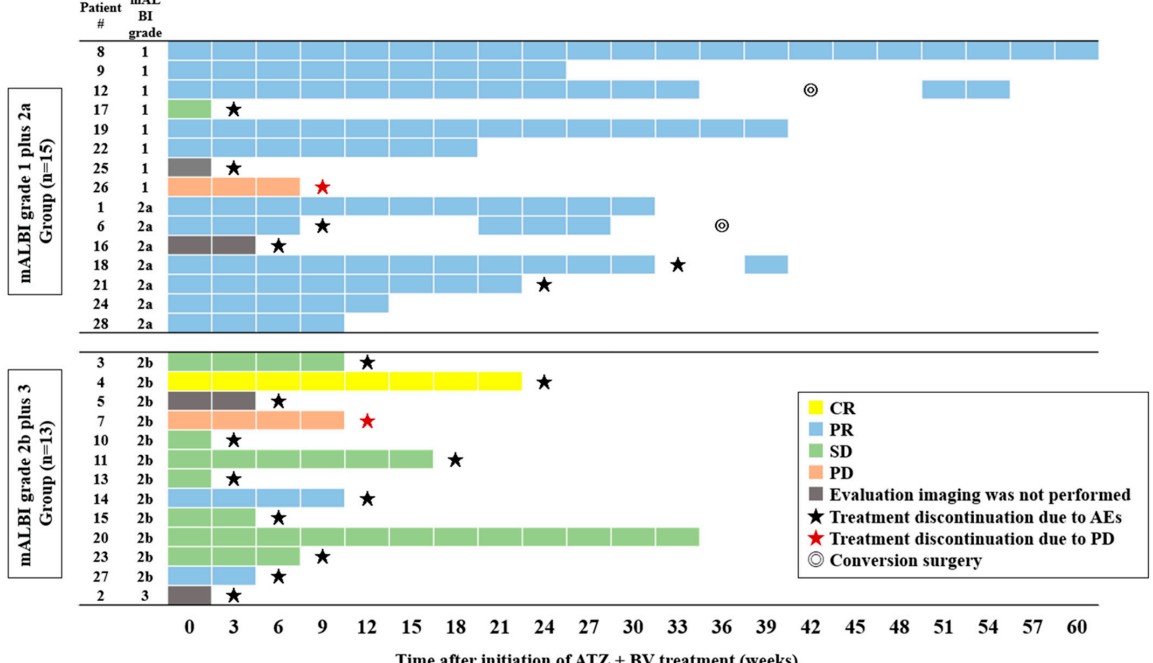

**Figure 3.** The clinical course of all patients treated with ATZ + BV, who were divided into mALBI grade 1 plus 2a ($n$ = 15) and 2b plus 3 groups ($n$ = 13). Abbreviations: AE—adverse event; CR—complete response; PR—partial response; SD—stable disease; PD—progressive disease; ATZ—atezolizumab; BV—bevacizumab.

*3.5. Safety and AEs*

The AEs that occurred during the treatment are summarized in Table 4. The most common AEs were pyrexia (12 cases, 42.9%), followed by fatigue (nine cases, 32.1%) and liver dysfunction and increased thyroid-stimulating hormone levels (TSH, each event accounted for seven cases [25%]), and renal dysfunction (six cases, 21.4%). Regarding the AEs of CTCAE version 5.0 grade three or higher, fatigue was observed in five cases, and liver dysfunction, rash, proteinuria, acute adrenal insufficiency, hemophagocytic syndrome, oral mucositis, decreased appetite, hypertension, and diarrhea were also observed. Among these severe AEs, liver dysfunction, rash, acute adrenal insufficiency, hemophagocytic syndrome, and oral mucositis were considered immune-related AEs (irAEs) that were treated with steroid therapy after the discontinuation of ATZ + BV treatment; all patients recovered promptly.

**Table 4.** Adverse events associated with ATZ + BV treatment.

| Adverse Events | Grade 1 or 2 | Grade 3 | Any Grade (%) |
|---|---|---|---|
| Pyrexia | 12 | 0 | 12 (42.9) |
| Fatigue | 4 | 5 | 9 (32.1) |
| Increased TSH | 7 | 0 | 7 (25.0) |
| Liver disfunction | 6 | 1 * | 7 (25.0) |
| Renal disfunction | 6 | 0 | 6 (21.4) |
| Rash | 4 | 1 * | 5 (17.9) |
| Proteinuria | 3 | 1 | 4 (14.3) |
| Decreased appetite | 3 | 1 | 4 (14.3) |
| Hypertension | 3 | 1 | 4 (14.3) |
| Stomatitis | 3 | 0 | 3 (10.7) |
| Ascites | 3 | 0 | 3 (10.7) |
| Edema | 3 | 0 | 3 (10.7) |
| Diarrhea | 2 | 1 | 3 (10.7) |
| Dicreased TSH | 2 | 0 | 2 (7.1) |
| Palmar–plantar erythrodysesthesia | 2 | 0 | 2 (7.1) |
| Oral mucositis | 0 | 2 * | 2 (7.1) |
| Heart failure | 1 | 0 | 1 (3.6) |
| Infusion reaction | 1 | 0 | 1 (3.6) |
| Acute adrenal insufficiency | 0 | 1 * | 1 (3.6) |
| Hemophagocytic syndrome | 0 | 1 * | 1 (3.6) |

Values are expressed as a number in grade 1 and 2 and grade 3 categories. Values are expressed as number (%) in any grade category. * These cases were treated with steroid therapy due to severe irAEs. TSH—thyroid-stimulating hormone; ATZ—atezolizumab; BEV—bevacizumab; irAE—immune-related adverse event.

The AEs observed in all patients and the clinical course after discontinuation are shown in Table 5. AEs were not observed in two cases and pyrexia was observed in 12 (42.9%), which tended to be associated with the irAEs caused by an autoimmune system imbalance due to ATZ. Seventeen patients discontinued ATZ + BV treatment due to AEs, one restarted ATZ + BV, three were switched to MTA, and five underwent BSC after discontinuation.

**Table 5.** Adverse events associated with ATZ + BV treatment in all cases.

| Case Number | Adverse Events | Pyrexia | Discontinuation | After Discontinuation |
|---|---|---|---|---|
| 1 | Liver disfunction * | No | No | |
| 2 | Fatigue, infusion reaction * | Yes | Yes | BSC |
| 3 | Fatigue, edema | No | Yes | BSC |
| 4 | Acute adrenal insufficiency *, increased TSH *, renal disfunction, fatigue | Yes | Yes | Observation |
| 5 | Fatigue, renal disfunction | No | Yes | BSC |
| 6 | Liver disfunction *, renal disfunction, stomatitis *, edema | Yes | Yes | ATZ + BV |
| 7 | Liver disfunction *, decreased TSH *, decreased appetite, fatigue | Yes | No | |
| 8 | Rash * | No | No | |
| 9 | Fatigue, proteinuria, increased TSH *, renal disfunction | No | No | |
| 10 | Diarrhea, ascites, liver disfunction *, proteinuria | Yes | Yes | MTA |
| 11 | Heart failure, proteinuria | No | Yes | MTA |
| 12 | None | No | No | |
| 13 | Rash *, increased TSH * | Yes | Yes | Observation |
| 14 | Oral mucositis *, rash *, stomatitis *, increased TSH * | No | Yes | Observation |
| 15 | Fatigue, decreased appetite, stomatitis *, palmar–plantar erythrodysesthesia * | Yes | Yes | BSC |
| 16 | Proteinuria, renal disfunction, hypertension, increased TSH * | No | Yes | Observation |
| 17 | Hemophagocytic syndrome *, liver disfunction * | Yes | Yes | MTA |
| 18 | Fatigue, decreased appetite, liver disfunction * | No | Yes | ATZ + BV |
| 19 | Rash, renal disfunction, liver disfunction * | No | No | |
| 20 | Decreased TSH * | No | No | |
| 21 | Ascites, increased TSH * | No | Yes | Observation |
| 22 | Fatigue, hypertension | Yes | No | |
| 23 | Rash *, palmar–plantar erythrodysesthesia *, increased TSH * | Yes | Yes | Observation |
| 24 | Hypertension | No | No | |
| 25 | Rash *, fatigue, decreased appetite | Yes | Yes | BSC |
| 26 | Diarrhea | No | No | |
| 27 | Oral mucositis * | Yes | Yes | Observation |
| 28 | None | No | No | |

* The adverse event especially related with immune checkpoint inhibitor. BSC—best supportive care; MTA—molecular target agent; TSH—thyroid-stimulating hormone; ATZ—atezolizumab; BEV—bevacizumab.

## 4. Discussion

This study demonstrated that the pretreatment mALBI grades were the important predictive factors associated with the therapeutic response and the treatment continuation of ATZ + BV therapy for patients with u-HCC, in the initial clinical experience. The results of the IMbrave150 trial reported that ATZ + BV treatment showed a therapeutic response against sorafenib in systemic chemotherapy for u-HCC, and demonstrated that various AEs, such as irAEs, were observed. The IMbrave150 trial excluded patients with Child–Pugh class B, prior systemic chemotherapy, and BCLC stage A or B. However, in a real-world clinical setting, some patients with early or intermediate stage HCC cannot undergo surgical treatment, locoregional treatment, or transcatheter arterial chemoembolization (TACE) due to their tumor's location, tumor condition, and other complications. Therefore, systemic therapy may be indicated for such patients with HCC, who have BCLC stage A or B. Moreover, reliable predictive markers of the therapeutic response are necessary for patients with u-HCC, treated by ATZ + BV therapy [22].

This study investigated the pretreatment factors associated with the therapeutic response and treatment continuation in a real-world setting. Consequently, good hepatic function, particularly in mALBI grades 1 and 2a, were the important predictive factors for both the therapeutic response and the treatment continuation of ATZ + BV treatment. On the other hand, several important clinical factors, including age, naïve or recurrent HCC, BCLC staging classification, and tumor markers, were not associated with either the therapeutic response or the treatment continuation. The importance of the mALBI grade has already been reported by Hiraoka et al., particularly for patients with u-HCC who were treated with lenvatinib, and it has been recently regarded as more important than the Child–Pugh classification in HCC chemotherapy [30]. Since the Child–Pugh classification is determined using both objective (ascites and hepatic coma) and semiquantitative factors and has not been established based on a statistical method, the mALBI grades are more suitable for HCC treatment than the Child–Pugh classification [25]. Therefore, ATZ + BV treatment is recommended for patients with u-HCC, with good hepatic function and with mALBI grades 1 and 2a, to achieve a therapeutic response.

Several articles have reported the factors associated with a therapeutic response in initial clinical experiences. Chuma et al. reported that ATZ + BV treatment might offer significant benefits in patients who meet the IMbrave150 trial or have a low NLR [31]. Eso et al. reported that pretreatment NLR might be a useful predictive factor associated with the therapeutic response [32]. NLR is a marker of systemic inflammatory response and reflects the balance between the neutrophils and lymphocytes [33]. NLR may also represent the balance between the pro-tumoral inflammatory status and the anti-tumoral immune response. Several reports have shown that low pretreatment NLR values are effective for antitumor effects in the immune checkpoint inhibitor (ICI) treatment for several types of cancer [34,35]; their efficacy also applies to HCC. Particularly, Hung et al. showed that patients with HCC with an NLR of 2.5 prior to ICI treatment had a better chance of disease control than those with an NLR higher than 2.5 [36]. However, the NLR cut-off value differs depending on each facility; therefore, attention to this should be paid when interpreting previous studies' results. This study showed no significant difference between the pretreatment NLR and the therapeutic response. To understand the other factors associated with the therapeutic response, Sho et al. reported that portal vein tumor thrombosis or the hepatitis B virus was significantly associated with PD ($p = 0.039$, $p = 0.050$) [17]. However, in our study, it was unclear whether PD increased in the case of vascular infiltration because the number of patients with PD was small.

Regarding treatment discontinuation, Chuma et al. reported that the frequency of treatment interruption due to fatigue was higher in patients with Child–Pugh class B, than in the patients with Child–Pugh class A [31]. However, the numbers of AEs leading to treatment interruption and/or withdrawal were not significantly different between patients with Child–Pugh class A and those with Child–Pugh class B. Moreover, the mechanism and overall frequency of the interruptions of, or withdrawal from ATZ + BV treatment associated

with AEs were not significantly different regarding the Child–Pugh classification, the MTA history, and the satisfaction of the IMbrave150 trial inclusion criteria [31]. Nevertheless, the present study revealed that patients with u-HCC, with good hepatic function could continue ATZ + BV treatment longer than those without; however, the causal relationships between pretreatment hepatic function and treatment continuation or the appearance of AEs are unclear.

There were no new safety concerns compared to those in the IMbrave150 trial. In this study, the most common AEs were pyrexia (42.9%), followed by fatigue (32.1%) and liver dysfunction (25.1%), increased TSH levels (25.1%), renal dysfunction (21.4%), and a rash (17.9%). Patients also experienced acute adrenal insufficiency, oral mucosal damage, and hemophagocytic syndrome, as rare and severe AEs, with CTCAE grade 3. These severe AEs were diagnosed as irAE in collaboration with other clinical departments; consequently, they were resolved by early steroid treatment, and the patients recovered gradually. In our study, severe irAEs were observed with pyrexia, which may be an important symptom that is suggestive of severe irAEs in the ATZ + BV treatment for u-HCC.

Several previous studies have reported AEs associated with ATZ + BV treatment. Iwamoto et al. reported that there was no significant difference in the frequency of AEs at any grade in MTA-naïve (94.7%) and MTA-experienced (100%) cases ($p = 0.15$). The frequencies of grade three AEs in MTA-naïve and MTA-experienced cases were 26.3% and 31.2%, respectively. There was no significant difference in the AEs of grade three between both groups ($p = 0.70$) [15]. Sho et al. reported that the safety profile was similar between patients who satisfied and did not satisfy the inclusion criteria for the IMbrave150 trial [17]. In this study, we could not investigate aspects regarding the condition of the IMbrave150 trial and pretreatment MTA because of the small sample size. Since the AEs associated with ATZ + BV treatment, especially irAEs, are one of the most important factors for sustaining the therapy, further analyses are required to confirm the safety profiles of patients treated with ATZ + BV therapy.

Our study had several limitations. First, this was a single-center study, with a small sample size. Therefore, additional studies are required. Second, the observation period was very short. Moreover, we could not evaluate post-treatment factors, such as changes in tumor markers, hepatic function, and the CT and MRI imaging characteristics after treatment. This was due to the short observation period and the small number of cases, making it difficult to evaluate the characteristics of the imaging. A more definitive conclusion requires a longer observation period and more cases to investigate the factors associated with the therapeutic response and treatment continuation.

## 5. Conclusions

This retrospective study assessed ATZ + BV treatment for patients with u-HCC, with initial experience in a real-world setting, and several important findings were obtained. Pretreatment mALBI grades were important factors that associated with the therapeutic responses and the treatment continuation. During the ATZ + BV treatment period, patients with mALBI grades 1 and 2a are expected to have successful treatment; however, careful monitoring is necessary for patients with mALBI grade 2b, for both the therapeutic response and treatment continuation. Further research is needed to increase the number of enrolled patients and to collect long-term follow-up data.

**Author Contributions:** T.T.: conceptualization, formal analysis, acquisition of data, and writing original draft. K.T., K.Y., H.F., R.Y., A.F. and S.S. (Shotaro Sakisaka): acquisition of data. S.S. (Satoshi Shakado) and F.H.: acquisition of data, and study supervision and editing. All authors have read and agreed to the published version of the manuscript.

**Funding:** This research received no external funding.

**Institutional Review Board Statement:** Approval for this retrospective study was obtained from the local ethical review board (approval No. H21-10-0002) of Fukuoka University Faculty of Medicine.

**Informed Consent Statement:** Written informed consent was obtained from the patients included in this study.

**Data Availability Statement:** The data of this study are available on request from the authors.

**Acknowledgments:** The appreciate all the patients who participated in this study and their families. We are grateful to all the investigators, physicians, nurses, pharmacists, and radiologists who assisted us with this study.

**Conflicts of Interest:** The authors declare no conflict of interest related to this study.

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
