# Peer review of "Pretreatment Modified Albumin–Bilirubin Grade Is an Important Predictive Factor Associated with the Therapeutic Response and the Continuation of Atezolizumab plus Bevacizumab Combination Therapy for Patients with Unresectable Hepatocellular Carcinoma"

_curroncol, doi:10.3390/curroncol29070381_

Round 1
Reviewer 1 Report
I think the sample size is still not enough (from 21 to 28 patients) to
evaluate efficacy of mALBI grade to predict OS after Aze/beva Tx for
advanced HCC.
Multivariate analysis is required to exclude confounding factors, thus
more than 100 patients (or at least 50 patients) are required.
Author Response
Response to Reviewer 1 Comments
I think the sample size is still not enough (from 21 to 28 patients) to evaluate efficacy of mALBI grade to predict OS after Aze/beva Tx for advanced HCC.
Multivariate analysis is required to exclude confounding factors, thus more than 100 patients (or at least 50 patients) are required.
Response:
Dear reviewer
Thanks for this insightful comment. We also appreciate the time taken to review this paper. As you indicated, we would prefer a sample size of 50 to 100 cases. However, increasing the sample size within the set study period was difficult. Interestingly, despite the increase in sample size from 21 to 28 cases, the important results remained the same. The conclusion was the same despite increasing the sample size from 21 to 28 patients: modified ALBI grade 1 and 2a was important for treatment response and continuation, and previous treatment for hepatocellular carcinoma and tumor stage were irrelevant. However, we understand that increasing the sample size is important to draw more detailed conclusions and will consider it for future studies.
Sincerely
The authors
Reviewer 2 Report
The study addresses a very important topic in HCC, since atezolizumab plus bevacizumab is part of everyday clinical practice following the practice-changing results of IMbrave150 phase III trial.
ICIs have revolutionized the treatment of several solid tumors. The mechanism of action of these anticancer agents acts on different pathways involved in tumor immune escape which is involved in tumor growth. Programmed cell death 1 (PD-1) and cytotoxic T-lymphocyte associated protein 4 (CTLA-4), with their ligands (PD-L1/2 and B7-1/2 respectively), play a pivotal role in this process and represent the main targets of several ICIs. Tumor immune escape has a central role also in HCC, as it usually arises in the context of chronic liver inflammation which promotes an immune exhausted microenvironment.
Based on these premises, the study addresses a timely topic in HCC.
Some changes are required in my opinion:
1. The background of HCC immunotherapy should be better discussed in the introduction and discussion sections, and some recent papers added, only for a matter of consistency (PMID: 29968763 ; PMID: 34429006). We think it could be important to introduce the topic of this interesting study exploring a very important subject.
2. A linguistic revision is necessary
3. The authors should expand the discussion section, including a more personal perspective to reflect upon.
4. the limitations of the current study should be further highlighted.
Major changes are required.
Author Response
Response to Reviewer 2 Comments
The study addresses a very important topic in HCC, since atezolizumab plus bevacizumab is part of everyday clinical practice following the practice-changing results of IMbrave150 phase III trial.
ICIs have revolutionized the treatment of several solid tumors. The mechanism of action of these anticancer agents acts on different pathways involved in tumor immune escape which is involved in tumor growth. Programmed cell death 1 (PD-1) and cytotoxic T-lymphocyte associated protein 4 (CTLA-4), with their ligands (PD-L1/2 and B7-1/2 respectively), play a pivotal role in this process and represent the main targets of several ICIs. Tumor immune escape has a central role also in HCC, as it usually arises in the context of chronic liver inflammation which promotes an immune exhausted microenvironment.
Based on these premises, the study addresses a timely topic in HCC.
Some changes are required in my opinion:
Dear reviewer
We would like to thank you for considering our manuscript for publication. We appreciate the time taken to provide valuable comments to improve the readability of our contribution to literature. We have diligently provided point-wise responses to all comments below.
- The background of HCC immunotherapy should be better discussed in the introduction and discussion sections, and some recent papers added, only for a matter of consistency (PMID: 29968763 ; PMID: 34429006). We think it could be important to introduce the topic of this interesting study exploring a very important subject.
Thanks a lot for this important suggestion. In accordance with your comment, we cited the article recommended, PMID: 29968763, as a suggestive case report of immunotherapy as neoadjuvant chemotherapy for the patient with hepatocellular carcinoma in the introduction section. Next, we cited the article (PMID: 34429006) in relation to the importance of elucidating the predictors of treatment response to immunotherapy in the introduction and discussion sections. We believe that both articles recommended were significant papers that deepened the understanding of the topic of this study.
- A linguistic revision is necessary
Thanks for your helpful comment. We have employed the service of Editage, a professional English editing company (https://www.editage.jp/), to correct the language problems in the revised manuscript.
- The authors should expand the discussion section, including a more personal perspective to reflect upon.
Thanks for your insightful comment. We have expounded the discussion section following your suggestion. However, we were concerned that the discussion might become ambiguous because of the points raised by other reviewers; we have made revisions and additions so that the content will not differ significantly.
- the limitations of the current study should be further highlighted.
Major changes are required.
Thanks for your helpful comment. We have highlighted in the study “limitation” that a more definitive conclusion requires a longer observation period and more cases to investigate the factors associated with therapeutic response and treatment continuation.
Sincerely
The authors
Reviewer 3 Report
I would like to thank you for the opportunity of reviewing this interesting paper that is focused on a very remarkable and challenging topic that is a lively argument also in the daily clinical practice.
Recently, systemic therapeutic strategies have been developed for patients with unresectable HCC. In particular, in the IMbrave150 trial, Atezolizumab (ATZ) + Bevacizumab (BV) treatment maintained the quality of life of patients with unresectable HCC and improved survival benefits more than sorafenib did. However, since HCC patients with a history of MTA therapy and those with Child-Pugh class B cirrhosis were excluded from the IMbrave150 trial, recent reports regarding the safety and efficacy of ATZ + BV in these patients are inadequate. The present study aimed to assess the efficacy and safety of ATZ + BV therapy for patients with u-HCC from the viewpoint of their clinical features and to evaluate the pretreatment factors related to the therapeutic response and continuation of ATZ + BV therapy in a real-world setting.
Papers that explore in depth this theme, that always represented a great challenge for all clinicians, but secondarily also for hepatologists, especially in the era of tailored medicine, could surely be of interest for this important journal. Moreover, this paper demonstrates the aim of finding objective and practical conclusions from the many studies that have been conducted in recent years.
This paper is pleasurable to read, although it suffers from minor limitations that Authors can easily adjust in order to slightly improve their review making it more eligible for this important Journal. Furthermore, Authors can improve some section of the paper, adding information and including other important references about this topic that, in my opinion, should be cited and discussed.
SPECIFIC COMMENTS
This manuscript is pleasurable and flowing; no major issues are appreciable.
Although language used is appropriate, I (I am not a native English speaker) recommend to Authors to obtain a certified native speaker with proficiencies in the scientific-medical field to complete properly this paper (if not jet done). Moreover, I recommend making a further revision of the manuscript to fix some small typing/language errors.
TITLE
The title is clear and focused on the results. However, in my opinion, it may result too long, thus I suggest shortening it. For example: “Pretreatment modified albumin-bilirubin grade is an important predictive factor associated with predict therapeutic response and continuation of atezolizumab plus bevacizumab combination therapy for patients with unresectable hepatocellular carcinoma”.
ABSTRACT
The abstract is well structured in all its section and, therefore, it properly reflects the main text highlighting the most important aspects of this paper.
In the “conclusion” sections: “This study demonstrated that pretreatment mALBI grades 1 and 2a were the important predictive factors..”. The sentence sounds a bit odd, I suggest to modify it in “This study demonstrated that pretreatment mALBI grades 1 and 2a were the important predictive factors..”
KEYWORDS
Authors did not correctly reported keywords from MeSH Browser. In particular, I checked for example “modified albumin-bilirubin grade” and “immune-related adverse events” on MeSH Browser and they were not KW. This is important, in my personal opinion, in order to increase the traceability of this paper (and consequently the possibility of the Journal to be cited by Readers and Stakeholders). I suggest the check of all KW.
INTRODUCTION
Although the introduction appears concise and interesting to read, I believe that the authors could enrich this section adding some important clarifications. In fact, when concepts are clearly and exhaustively explicated readers may become more passionate about reading the paper, thus increasing the chances to using it as a reference.
In my opinion, it is very important to expand the section regarding the available therapies for unresectable HCC and what are the current options and limits in the second line and beyond, especially now that Atezolizumab+Bevacizumab has been approved as the first-line setting. [Role of immunotherapy in the management of hepatocellular carcinoma: current standards and future directions. Curr Oncol. 2020 Nov;27(Suppl 3):S152-S164. doi: 10.3747/co.27.7315] [Management of hepatocellular carcinoma after progression on first-line systemic treatment: defining the optimal sequencing strategy in second line and beyond. Curr Oncol. 2020 Nov;27(Suppl 3):S173-S180. doi: 10.3747/co.27.7103] In particular, a reasonable second-line treatment would be a tyrosine-kinase inhibitor (tki), either Lenvatinib or Sorafenib, with Lenvatinib standing out as the one with the highest reported response rate and longest pfs. Third-line treatment would consist of a second-line tki such as cabozantinib or regorafenib.
Moreover, could the Authors please discuss more exhaustively why there is a current need assess the efficacy and safety of ATZ + BV therapy in a real-world setting? Even if these considerations are elucidated in the Discussion section, I believe that briefly reporting them also in the Introduction section could further highlight the importance of the present study and engage readers’ interest.
MATERIALS & METHODS
In the section 2.1. “Patient population”: “HCC was diagnosed by radiological imaging using contrast-enhanced computed tomography (CT) or magnetic resonance imaging (MRI) combined with serum tests, particularly tumor markers, such as alpha-fetoprotein (AFP) and des-gamma carboxy prothrombin (DCP)” What imaging features did you consider for diagnosis? Were CT and MRI performed according to the standards of reference recommended by international guidelines? [European Association for the Study of the Liver. EASL Clinical Practice Guidelines: Management of hepatocellular carcinoma. J Hepatol. 2018 Jul;69(1):182-236. doi: 10.1016/j.jhep.2018.03.019].
In the section 2.2. “Study design”: “Therapeutic responses were determined using both the Response Evaluation Criteria in Solid Tumors (RECIST) version 1.1, and modified RECIST (mRECIST) guidelines”. It is well known that the decision on whether a patient is a responder or progressor after treatment may vary among different operators, especially in case of a non-specifically trained radiologist and, therefore, regardless of the adopted criteria, patients should be evaluated by experienced radiologists to minimize variability in this critical instance [Eur Radiol. 2018;28(9):3611-3620. doi:10.1007/s00330-018-5393-3]. Please, could the Authors report the methodology used (an experienced radiologist evaluated CT?) citing the aforementioned paper [Eur Radiol. 2018;28(9):3611-3620. doi:10.1007/s00330-018-5393-3].
RESULTS
Results remain clear and well-structured, and no further adjustments are needed.
DISCUSSION
Discussion is well structured and well-presented.
However, I suggest expanding the limitations sections.
Morever, I would suggest adding some considerations regarding the introduction of new diagnostic algorithms for HCC in the near future [Proposal of a new diagnostic algorithm for hepatocellular carcinoma based on the Japanese guidelines but adapted to the Western world for patients under surveillance for chronic liver disease. J Gastroenterol Hepatol. 2016 Jan;31(1):69-80. doi: 10.1111/jgh.13150.]. These diagnostic algorithms, in fact, will allow to identify smaller and smaller lesions and therefore also in earlier stages. Do the Authors think that molecular target agent therapy will still have an important role in these future scenarios and what could possibly imply the reduction of its execution in terms of costs?
Finally, a final clinical implication of the present study was that these prognostic results could also be used as a guide to choose the best imaging technique to adopt for the follow-up. Taking into account the different peculiarities of the imaging methods (CT and MRI) recommended by the EASL guidelines to assess the treatment response, it would be possible to tailor the best imaging modality case by case. In fact, in future, in the absence of a satisfactory therapeutic response with the need of treatment continuation, it will be preferable to perform MRI instead of CT due to its better diagnostic accuracy in the assessment of the therapy response. Furthermore, starting from a management point of view, a test that predicts the chance of response (or not) allows planning the clinical/radiological management. [TRANS-TACE: Prognostic Role of the Transient Hypertransaminasemia after Conventional Chemoembolization for Hepatocellular Carcinoma. J Pers Med. 2021 Oct 17;11(10):1041. doi: 10.3390/jpm11101041]
TABLES & FIGURES
Table 1 should be re-formatted. For example, adding some lines could help readers to better understand the presented data.
The remanent Tables and Figures are satisfactory and they correctly match the quality standard of this Journal.
REFERENCES
References do not reflect the style showed in “Authors Guidelines”. (Author 1, A.B.; Author 2, C.D. Title of the article. Abbreviated Journal Name, Year, Volume, page range). Please, format them accordingly.
Round 2
Reviewer 2 Report
Acceptance.
Author Response
Dear reviewer;
We would like to thank you for considering our manuscript for publication. We appreciate the time taken to provide valuable comments.
Sincerely,
The authors
This manuscript is a resubmission of an earlier submission. The following is a list of the peer review reports and author responses from that submission.
Round 1
Reviewer 1 Report
The paper by Tanaka et al. demonstrated that pretreatment mALBI grades 1 and 2a were the im-28 portant predictive factors associated with therapeutic response and therapeutic continuation of 29 ATZ + BV for patients with u-HCC.
However, due to the small number of patients enrolled, they shoud increase the number of patients enrolled and collect long-term follow up data, additional studies are required to draw definitive conclusions.
Author Response
The paper by Tanaka et al. demonstrated that pretreatment mALBI grades 1 and 2a were the important predictive factors associated with therapeutic response and therapeutic continuation of ATZ + BV for patients with u-HCC.
However, due to the small number of patients enrolled, they should increase the number of patients enrolled and collect long-term follow up data, additional studies are required to draw definitive conclusions.
Response:
Dear reviewer
Thanks for this insightful comment. We agree that the number of enrolled patients should be increased and long-term follow-up data be collected. We have now acknowledged this and suggested it as a topic for further research in the Conclusion section of the revised manuscript (Page12, line351-352). We adapted our manuscript to answer your concerns. All changes are marked in track changes in the document attached. Please note that we received feedback from other reviewers as well, which are also reflected in these changes in the manuscript.
Reviewer 2 Report
It's simple and easy to understand.Author Response
Response:
Dear reviewer
Many thanks for your commendation of our work. We aimed to simplify the study design and concisely discuss the results. Please find attached the adapted manuscript, with all changes marked in track changes.
Sincerely
The authors
Reviewer 3 Report
General comment: in this study, the authors evaluate efficacy of mALBI score predicting tumor response after Ate/beva for uHCC. While this is a topic of great interest and new, I have following major concerns.
- Most importantly, sample size is too small to draw any meaningful conclusion. In addition, study patients are too heterogenous (naive HCC = 9, recurrent HCC: 11, BCLC A/B/C). I would recommend to gather more sample size.
- Why not do evaluate overall survival? OS is more important than tumor response.
Round 2
Reviewer 3 Report
General comment: this is revised version of previous manuscript. Although it is good topic and of interest, the sample size and study patients were still major limitations (naive HCC = 9, recurrent HCC = 11, BCLC A/B/C). Thus, it still cannot give any meaningful findings. This reviewer recommend to gather more sample size.